# Biocrusts buffer against the accumulation of soil metallic nutrients induced by warming and rainfall reduction

Eduardo Moreno-Jiménez [1✉], Raúl Ochoa-Hueso[2], César Plaza [3], Sara Aceña-Heras[1], Maren Flagmeier[1,4], Fatima Z. Elouali[5], Victoria Ochoa[6], Beatriz Gozalo[6], Roberto Lázaro[7] & Fernando T. Maestre [6,8]

The availability of metallic nutrients in dryland soils, many of which are essential for the metabolism of soil organisms and vascular plants, may be altered due to climate change-driven increases in aridity. Biocrusts, soil surface communities dominated by lichens, bryophytes and cyanobacteria, are ecosystem engineers known to exert critical functions in dryland ecosystems. However, their role in regulating metallic nutrient availability under climate change is uncertain. Here, we evaluated whether well-developed biocrusts modulate metallic nutrient availability in response to 7 years of experimental warming and rainfall reduction in a Mediterranean dryland located in southeastern Spain. We found increases in the availability of K, Mg, Zn and Na under warming and rainfall exclusion. However, the presence of a well-developed biocrust cover buffered these effects, most likely because its constituents can uptake significant quantities of available metallic nutrients. Our findings suggest that biocrusts, a biotic community prevalent in drylands, exert an important role in preserving and protecting metallic nutrients in dryland soils from leaching and erosion. Therefore, we highlight the need to protect them to mitigate undesired effects of soil degradation driven by climate change in this globally expanding biome.

[1] Department of Agricultural and Food Chemistry, Universidad Autónoma de Madrid, 28049 Madrid, Spain. [2] Department of Biology, IVAGRO, University of Cádiz, Campus de Excelencia Internacional Agroalimentario (ceiA3), Campus del Rio San Pedro, 11510 Puerto Real, Cádiz, Spain. [3] Instituto de Ciencias Agrarias, Consejo Superior de Investigaciones Científicas, Serrano 115 bis, 28006 Madrid, Spain. [4] Department of Biology (Botany), Universidad Autónoma de Madrid, 28049 Madrid, Spain. [5] Department of Agronomy, Faculty of Sciences of Nature and Life, University of Mascara, 29000 Mascara, Algeria. [6] Instituto Multidisciplinar para el Estudio del Medio "Ramon Margalef", Universidad de Alicante, Carretera de San Vicente del Raspeig, s/n 03690, San Vicente del Raspeig, Alicante, Spain. [7] Estación Experimental de Zonas Áridas Consejo Superior de Investigaciones Científicas, Carretera de Sacramento, s/n 04120La, Cañada de San Urbano, Almería, Spain. [8] Departamento de Ecología, Universidad de Alicante, Carretera de San Vicente del Raspeig, s/n 03690, San Vicente del Raspeig, Alicante, Spain. ✉email: eduardo.moreno@uam.es

Soil metallic nutrients, including Cu, Fe, K, Mg, Mn, Na and Zn, participate in critical processes such as cell redox homoeostasis and photosynthesis[1,2], and thus are essential for microbial and plant growth[3]. Likewise, these metallic nutrients also play key roles in biogeochemical processes[4] such as plant litter decomposition and N fixation[5–8]. Climatic conditions (e.g., aridity) and soil properties (e.g., pH, organic matter and clay content) may induce alterations in the availability of metallic nutrients[9], affecting the performance of soil organisms and plants, as well as the many processes that depend on them[10]. These alterations in metallic nutrient availability may, in turn, alter the supply of life-supporting ecosystem services like plant biomass and food production[4,11].

Currently, ~45% of the Earth's terrestrial surface is occupied by drylands[12]. These areas include hyper-arid, arid, semiarid and dry-subhumid ecosystems and collectively constitute the Earth's largest biome[13]. Their global extent and socio-ecological influence will increase due to projected increases in aridity associated with climate change[14]. The scarcity of water that characterises drylands worldwide limits plant growth, and thus the production of crops, forage, wood and other provisioning ecosystem services[15]. Soil fertility and nutrient cycling in drylands are constrained not only by low water availability, but also by the high pH values, low organic carbon contents and the low degree of weathering typical of soils in these areas[16]. Metal availability decreases as aridity increases in global drylands[9], a response that may limit even further the availability of metallic nutrients in a warmer and more arid world[14]. However, and despite the importance of metals for many metabolic processes and thus for sustaining plant and animal growth, their responses to realistic climate change scenarios in terrestrial ecosystems are largely unknown[9].

Biocrusts, which consist of associations of cyanobacteria, green algae, fungi, protists, bacteria, lichens and bryophytes living on the soil surface, are a key biotic component of dryland ecosystems worldwide[17]. In these environments, biocrusts are considered ecosystem engineers due to their ability to regulate biogeochemical nutrient and water cycles[18], increase soil fertility[19] and affect the establishment and performance of organisms such as vascular plants, nematodes and microarthropods[20,21]. Although biocrusts are important modulators of soil surface biogeochemistry and hydrology, very few experiments have explored the interaction of biocrust abundance (cover) with soil metallic nutrients. Previous studies showed positive relationships between biocrusts and the bioavailability of soil metallic micronutrients such as Zn and Mn[22,23], and some even suggested that biocrusts may improve Zn, Fe, Mn and Mg uptake by associated vascular plants[24,25]. Biocrusts are also highly affected by the short-term manipulation of climatic variables, and previous studies have shown detrimental effects of experimental warming and rainfall reductions in the cover of biocrust constituents such as mosses and lichens[26,27]. However, little is known about the impacts of climate change on the capacity of biocrusts to modulate the responses of soil metallic nutrients to altered temperature and rainfall regimes. Biocrusts have certainly potential for doing so given the roles that metal availability plays on the performance of biocrust constituents such as mosses and lichens[28–30] and the known role of biocrusts as modulators of the responses of soil C and N to climate change due to limited leaching/erosion and increased biophysical protection[31,32].

Despite the importance of biocrusts for maintaining ecosystem structure and functioning in drylands worldwide[33] and the implications of climate change-induced effects on soil metals in these regions[9], no previous study has, to our knowledge, evaluated how climate change-induced impacts on biocrusts affect the availability of soil metallic nutrients in drylands. To fill this gap, we carried out a 7-year manipulation experiment and assessed the effects of experimental warming (ambient vs. 2–3 °C increment), rainfall reduction (ambient vs. 33–36% exclusion) and biocrust cover (poorly vs. well-developed biocrusts) on soil metallic nutrient content and availability. Here, we aimed to explore: (1) the relationships between simulated climate change, soil properties and soil metal content and availability, and (2) whether biocrusts modulate these relationships, which is an important knowledge gap in current projections of climate change effects on dryland ecosystems. Based on previous observational work[9], we hypothesised that increases in aridity induced by warming and rainfall reduction will decrease metal availability in soils, and that these effects will be less evident under high biocrust cover due to the amelioration of microclimatic conditions and the increase in soil fertility generated by the presence of well-developed biocrusts[34].

## Results

**Occurrence and availability of soil metals as affected by biocrusts, warming and rainfall exclusion.** Total and available metal concentrations in soils were assessed at the beginning of the experiment (in 2010, before treatments) and after 7 years of applying treatments in the form of warming and rainfall reduction to plots with either low or high biocrust cover. On average, the total content and bioavailability of metallic nutrients, as well as the total organic C content and pH, in soils under initially high biocrust cover were similar or slightly greater than those under low biocrust cover both at the beginning (2010; Supplementary Figs. 1 and 2) and at the end (2017; Fig. 1 and Supplementary Fig. 3) of the experiment. The only exception was the content of available Na at the beginning of the experiment, which was smaller in soils under high biocrust cover than in those under low biocrust cover (Supplementary Fig. 2).

After 7 years of experiment, reduced rainfall combined with warmer temperatures increased the contents of available K ($t = 2.55$, $P = 0.021$, df = 16), Mg ($t = 1.83$, $P = 0.085$, df = 16), Na ($t = 2.81$, $P = 0.012$, df = 16) and Zn ($t = 1.63$, $P = 0.123$, df = 16) with respect to the ambient treatment in soils under low biocrust cover, but not in those under high biocrust cover (Fig. 1). The content of available Zn in soils under both low and high biocrust cover also increased with respect to the ambient treatment ($t = 2.32$, $P = 0.034$, df = 16; and $t = 3.94$, $P = 0.001$, df = 16, respectively) after 7 years of reduced rainfall (Fig. 1). Seven years of experimental warming consistently increased the content and availability of metallic macronutrients (Fig. 2). Similarly, the availability of Zn and Cu increased more under reduced rainfall than under ambient conditions.

**Influence of each treatment on soil metal availability.** Our structural equation models (SEMs) indicated that, in 2017, the observed greater availability of metallic nutrients in soils under high biocrust cover was in part due to their greater total contents (Fig. 3 and Supplementary Fig. 5). Rainfall exclusion exerted positive direct and indirect effects on Zn availability; the latter mediated through a weak positive effect on total content. The effects of warming on the availability of K and Mg were mostly direct. Regarding temporal changes in total and available fractions (Fig. 4 and Supplementary Fig. 6), and similar to the models obtained for the 2017 sampling campaign, greater biocrust cover was associated with higher availability of Fe and Zn through time due to greater accumulation of their total contents. In the case of K, we also found a negative, direct association of the presence of high initial biocrust cover. Most of the effects of warming on available nutrients were direct; we only observed an indirect effect of rainfall on nutrient availability in the case of Zn.

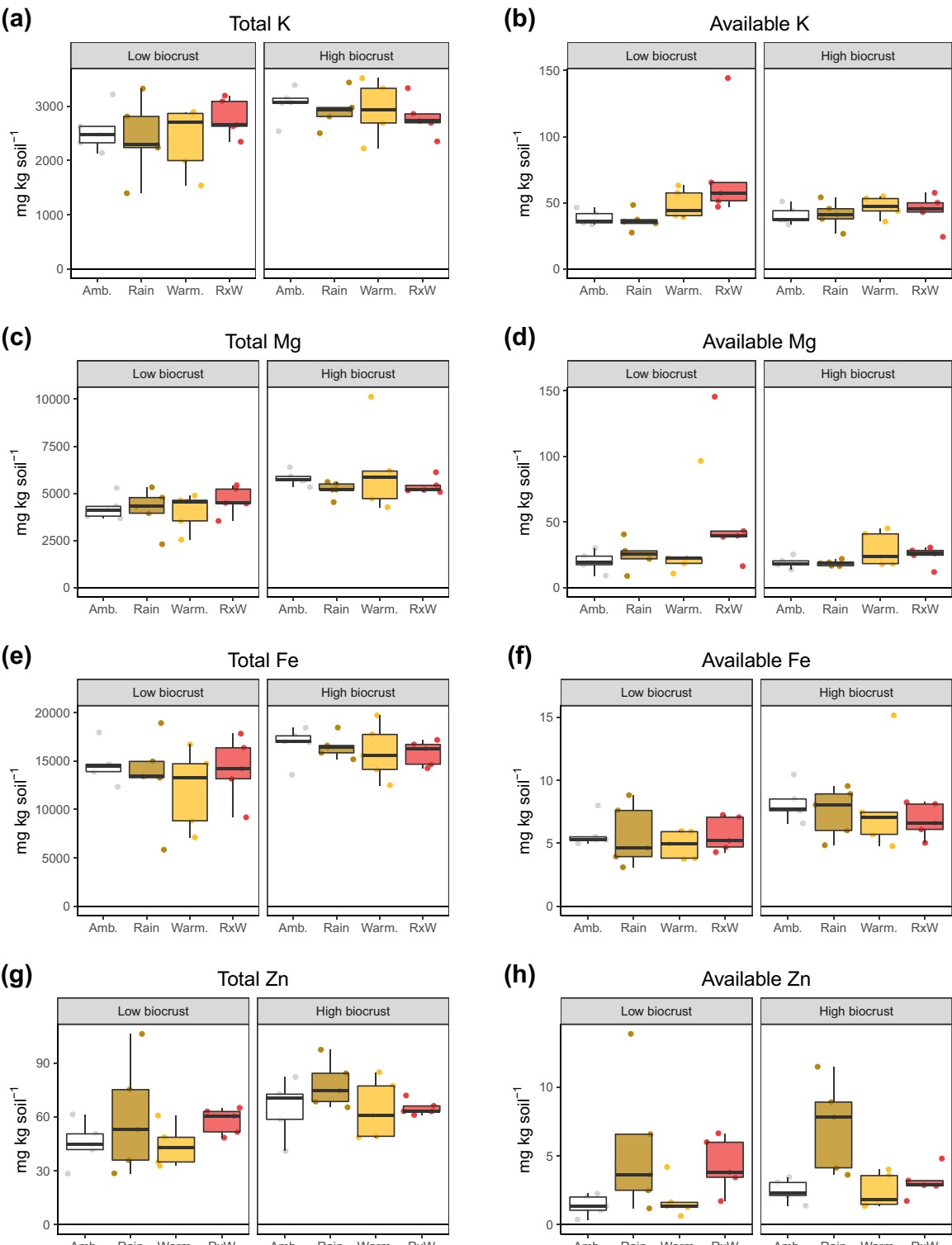

**Fig. 1 Total and available metal concentrations in soils.** Total K (**a**), Mg (**c**), Fe (**e**) and Zn (**g**) and available K (**b**), Mg (**d**), Fe (**f**) and Zn (**h**) in soils (mg kg$^{-1}$) under the different biocrust cover (low biocrust vs. high biocrust), warming and rainfall exclusion levels evaluated 7 years after the beginning of the experiment: control (Amb.), rainfall exclusion (Rain), warming (Warm.) and rainfall exclusion and warming (RxW) (box and whisker plots, $n = 5$).

## Discussion

Seven years of simulated climate change (reduced rainfall and warming) increased the availability of metallic nutrients in the soils of a semiarid grassland. This may be interpreted as a positive outcome at first, given the low values of these nutrients typically found in dryland soils[9]. However, this also indicates that metallic nutrients became more soluble under warming and reduced rainfall, and were therefore more susceptible to leaching and erosion, possibly leading to nutrient depletion over time. Our results clearly show that soils with a well-developed biocrust community are less sensitive and vulnerable to the effects of warmer and drier environmental conditions on metal nutrient

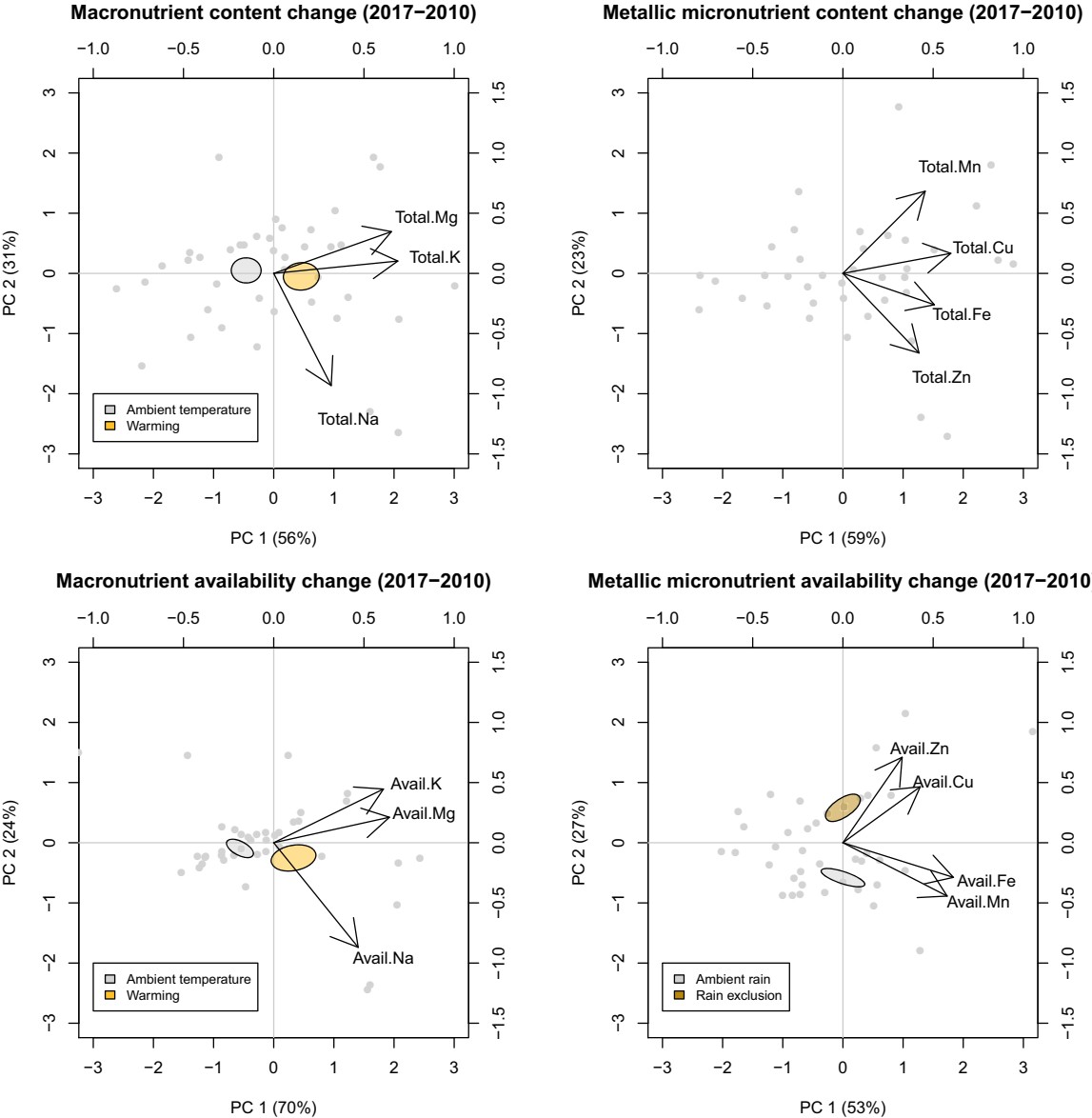

**Fig. 2 Results of principal component (PC) analyses.** Plots show PC analysis of the change from 2010 to 2017 in total and available contents of metallic macro and micronutrients. Ambient in blue, warming in orange and rainfall exclusion in red.

availability. This implies that any damage to biocrusts caused by human activities resulting in a reduction in their cover[32,35,36] will have far-reaching consequences in terms of the cycling and availability of metallic nutrients in soils that would also become more nutrient-impoverished.

Simultaneous rainfall reduction and warming resulted in clear-cut consequences in terms of the availability of K, Mg, Na and Zn 7 years after the beginning of the experiment, while responses to warming were less evident. The direct influence of soil pH on total soil metal contents may reflect the more marked presence of metal-containing basic minerals such as carbonates and/or phosphates with higher pH[37–40]. At the same time, metallic nutrient availability may increase with warming because many reactions resulting in metal solubilisation, such as mineral/salt dissolution and organic matter mineralisation, are related to temperature[41–44]. We attribute our findings to an increased solubility of mineral components, increased pH, and greater microbial activity due to mineralisation of dead soil organisms under drought, and also to the reduced leaching and plant nutrient uptake typically found under conditions of reduced water availability[45].

Previous studies describing the relationships of rainfall exclusion and warming with soil metallic nutrients are scarce and show unclear patterns. For example, a 6-year field experiment in a Mediterranean shrubland in the NE of Spain showed increases in total soil Fe and available Mg and Fe with rainfall reduction, but no effects on total Mg, Ca or Na[46] nor on total or extractable Zn or Cu[47]. Another field experiment in the a Chinese semiarid ecosystem revealed that 3 years of 1–2 °C warming increased total soil Mn and Cu, while a 2–3 °C warming increased the availability of Cu, Mn and Zn and decreased available Fe[48]. The results from incubation experiments showed that a decrease of soil humidity for 21 days was associated with greater solubilisation of soil Zn[49], but there were no clear patterns in Cu availability in response to increasing temperature after a 60-day incubation[50]. However, none of the studies mentioned above explored the role of soil biocrusts, a key biotic component in dryland ecosystems, in regulating soil metallic nutrient responses to warming and/or rainfall exclusion.

Higher levels of biocrust cover were associated with greater abundance or availability of several metallic nutrients (all the

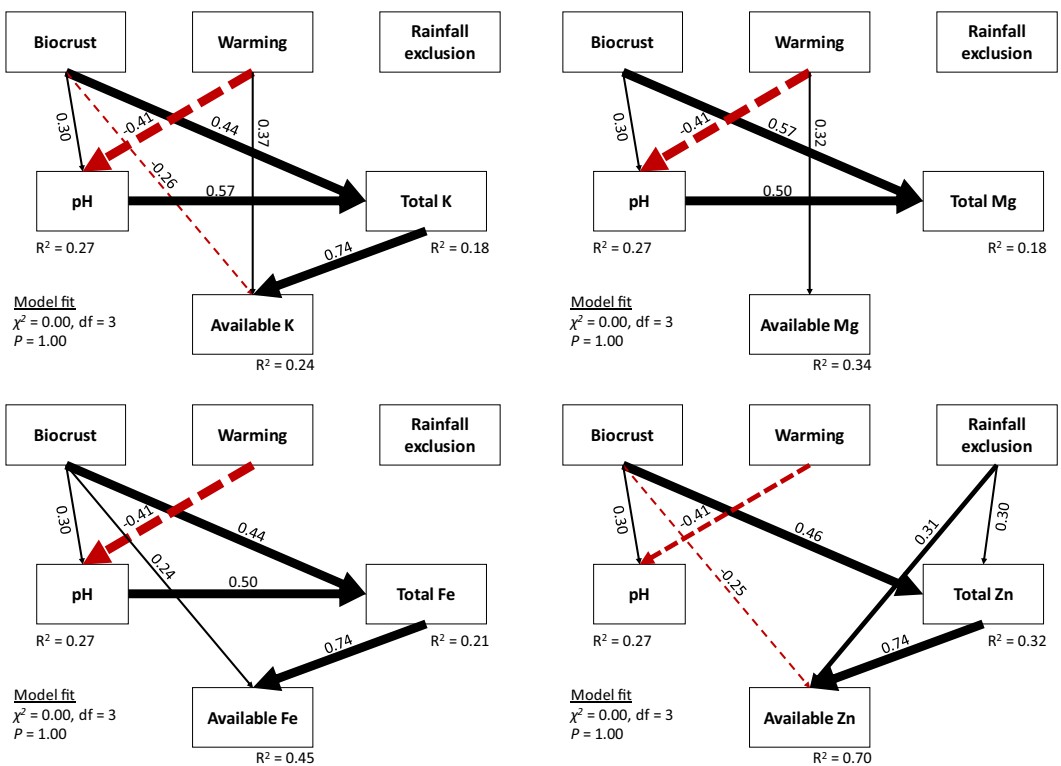

**Fig. 3 Structural equation models of biotic and abiotic effects in 2017.** Variables included are biocrusts, warming, rain exclusion and soil pH on total and available K, Mg, Fe and Zn. Measurements were taken in 2017, 7 years after the beginning of the experiment. Numbers adjacent to arrows are standardised path coefficients (analogous to relative regression weights) and indicative of the effect of the relationship. Continuous arrows show positive and dashed arrows negative relationships, with arrow thicknesses proportional to the strength of the relationship. The proportion of variance explained ($R^2$) is shown besides each response variable in the model. Goodness-of-fit statistics are shown in the lower left corner as the $\chi^2$. The a-priori model was refined by removing paths with coefficients close to zero (see the a-priori model in Supplementary Fig. 4).

elements examined except Na), highlighting the intimate relationship between these communities and soil metallic nutrients. While scarcity of metallic nutrients such as Zn and Mn can limit the growth of biocrust-forming lichens in drylands[22,28], biocrusts generally develop better in spots with high concentration of nutrients[28,51], and thus soil metal enrichment can be expected to favour biocrust growth and cover. Concomitantly, biocrusts may favour an enrichment of metallic nutrients in the soil by retaining efficiently fine soil and sediment particles[52], which are usually richer in metals than coarse fractions[53]. We found that the availability of Cu, Fe and Mn was higher under well-developed biocrusts regardless of the experimental treatment considered. Fungi like those forming part of the lichen symbiosis[54] are known to exudate organic acids that can form available complexes with these metallic micronutrients[55,56], which may also contribute to explain our results.

Plots with higher initial biocrust cover were less sensitive to the increase in the availability of some metallic nutrients (K, Mg, Zn and Na) induced by rainfall reduction and warming in our experiment (Figs. 1 and 3 and Supplementary Fig. 5). Biocrusts have been consistently shown to be able to modulate the storage and cycling of elements like C, N or P under changing climatic conditions[31,32,34,57], but up to now we did not have so strong evidences in relation to metallic nutrient cycling, especially Mg, Mn or Zn[22,28,51]. Our results suggest that complex interactions between climate change drivers and biocrusts regulate the response of metallic nutrients to altered climatic conditions. These communities form "islands of fertility" in the soils underneath them because they can create convenient conditions to support biological activity in drylands[23,34]. Such nutrient enrichment implies that the biota associated with biocrusts may

be able to use the greater available pool of metals induced by climate change, which would therefore reduce their accumulation in the soil and thus prevent their losses. In addition, key biocrust constituents like mosses and lichens control the hydrology of adjacent soils by increasing infiltration and increasing soil water storage after rainfall events, respectively[58,59], which can also influence the geochemistry of metals because the solubilisation of minerals requires the presence of water and the dynamics of wet-dry cycles changes soil metal speciation[60].

The outcomes of our study advance our previous understanding of the potential impacts of climate change on metal cycling in dryland ecosystems, which was based on observational evidence along aridity gradients and showed that increased aridity reduces the availability of metals in dryland soils worldwide[4,9]. In contrast, here we found that realistic simulations of rainfall reductions and warming cause the short-term accumulation of available metallic nutrients in the soil. Similar discrepancies have been observed when comparing soil nutrient responses to climate change obtained using manipulative experiments or observations gathered along climatic gradients[45]. Yuan et al.[45] have argued that manipulation experiments predict causal influence of warming/rainfall exclusion at the short-to-medium temporal scales (i.e., months or years), while environmental gradients give information that is more indicative of the expected effects in the long-term (i.e., centuries or millennia). According to this idea, the accumulation of soil metals under warming and reduced rainfall observed in our manipulative experiment may shift in the long-term because of losses via leaching and erosion, soil organic matter depletion and changes in soil pH 9[61–66]; yet a well-developed cover of biocrusts can palliate such losses by stabilising metallic nutrient cycles in the soil.

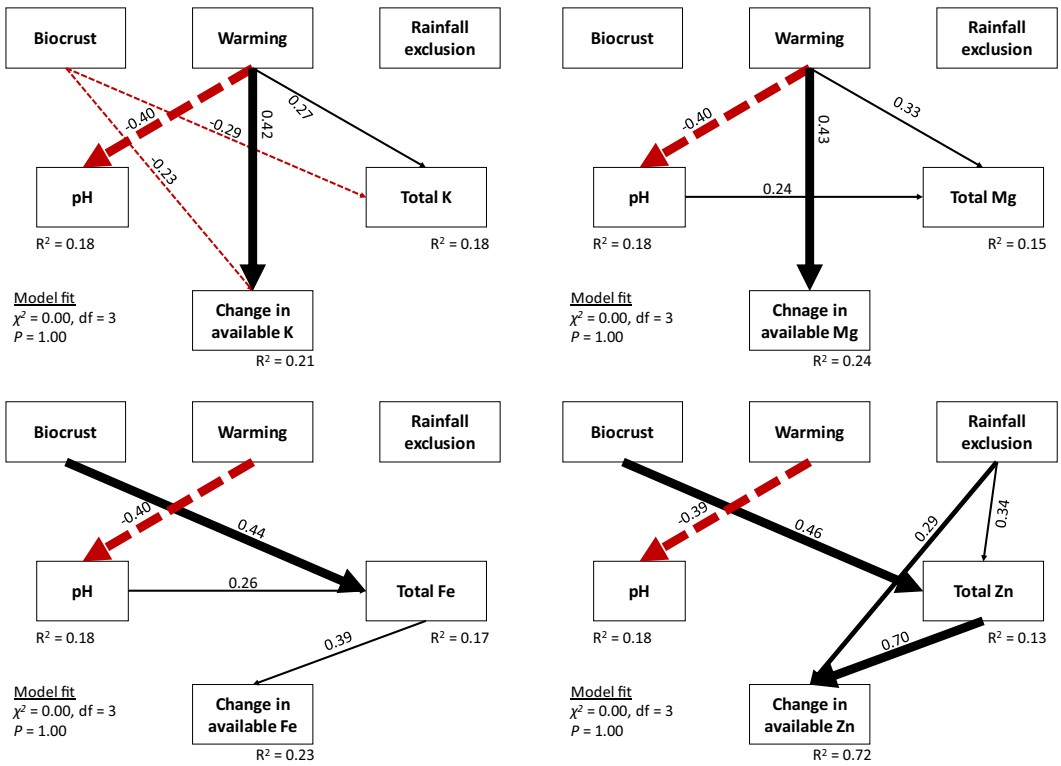

**Fig. 4 Structural equation models of biotic and abiotic effects during the experiment (2010–2017).** Variables included are biocrusts, warming, rain exclusion and soil pH on the change of total and available K, Mg, Fe and Zn. Measurements were taken during the experiment (2010–2017). Numbers adjacent to arrows are standardized path coefficients (analogous to relative regression weights) and indicative of the effect of the relationship. Continuous arrows show positive and dashed arrows negative relationships, with arrow thicknesses proportional to the strength of the relationship. The proportion of variance explained ($R^2$) is shown besides each response variable in the model. Goodness-of-fit statistics are shown in the lower left corner as the $\chi^2$. The a-priori model was refined by removing paths with coefficients close to zero (see the a-priori model in Supplementary Fig. 4).

## Conclusions

The combination of 2–3 °C warming and 33–36% rainfall reduction caused significant increases in soil metal availability (K, Mg, Na and Zn) in a biocrust-dominated semiarid ecosystem. Our results thus suggest that projections of current climate change may promote an accumulation of available metallic nutrients in dryland soils, which can be interpreted as a positive outcome. However, when analysed in the context of previously reported long-term effects of increasing aridity on metallic micronutrients[4,9], our data suggest that the observed accumulation may be a transient short-term effect, and that mobilisation may occur and result in a depletion of essential macro and micronutrients in the long term. Moreover, we showed for the first time, to our knowledge, that soils with the presence of a well-developed biocrust community were less sensitive to increases in metal availability under the simulation of realistic climate change scenarios. Therefore, promoting management strategies aimed at preserving, enhancing and/or even restoring biocrusts may be a suitable measure to increase the resistance of dryland soils to climate change.

## Methods

**Experimental design.** We used an experiment established in 2010 in Sorbas, Almería, South East Spain (37°05′N–2°04′W; 397 m a.s.l.), under a semiarid Mediterranean climate defined by dry and hot summers with a mean temperature of 17 °C and an annual precipitation of 270 mm. Soils are classified as Gypsiric Leptosols (IUSS Working Group WRB, 2006), have pH values ~7 and 1–12% of inorganic C (as carbonates). The site has a plant cover of <40% (dominated by *Stipa tenacissima*), and the open areas between plant patches contain well-developed biocrust formations dominated by lichens, i.e. *Diploschistes diacapsis*, *Fulgensia subbracteata*, *Squamaria lentigera*, *Buellia epipolia*, *Psora decipiens* or *Squamaria cartilaginea*[31].

The experiment consists of a full factorial design, with three factors, each with two levels and eight replicates per combination of treatments: (1) biocrust cover (<20% vs. >50%), (2) warming (control vs. 2–3 °C temperature increase), and (3) rainfall reduction (control vs. ~35% rainfall reduction). The biocrust cover changed dynamically during the 7 years of experimentation, but the treatments are named either "low" or "high" according to their biocrust cover status at the start of the experiment. Experimental plots are 1.2 × 1.2 m in size and were located at least 1 m apart. Temperature is experimentally manipulated with hexagonal open top chambers (OTCs) with sloping sides made of methacrylate (40 × 50 × 32 cm each) placed on the plots to trap heat inside the chamber. The OTCs also have the effects of decreasing soil moisture by ~1.5%[31]. Rainfall is experimentally reduced with rainout shelters made of methacrylate, which exclude 33–36% of the rain that falls in each rainfall event. In each plot, a PVC collar (20 cm diameter, 8 cm height) was inserted 5 cm into the soil for monitoring biocrust composition and cover. Further details about the experimental design can be found in Maestre et al.[31].

**Soil sampling and analyses.** Soil samples from the upper layer of soil (0–2 cm) were collected at the beginning of the experiment (early summer 2010) and 7 years later (early summer 2017). We focused on this layer because microbial biomass is generally highest in the first few centimetres of topsoil[67]. Once in the lab, visible biocrust pieces were removed from the soil, which was air-dried for one month and then sieved to <2 mm prior to analyses. Soil pH was determined in 1:5 soil: distilled water suspension. Available metals in soil were extracted following the Lindsay and Norvel[68] method, with a solution 0.005 M diethylenetriaminepentaacetic acid (DTPA), 0.01 M $CaCl_2$ and 0.1 M triethanolamine (pH 7.3), shaking 2 h at room temperature and filtering. Total metals were extracted with a mix $HNO_3/H_2O_2$, heating at 125 °C for 30 min and filtering[69,70]. The DTPA and total soil extracts were analysed by ICP-OES for metallic micronutrient (Cu, Fe, Mn and Zn) and macronutrient (K, Mg and Na) determination[68].

**Statistics and reproducibility.** This field experiment is a manipulative full factorial design. We sampled surface soils in five independent replicates (adjacent plots) for each combination of factors in 2010 and 2017. We analysed the univariate and interaction effects of initial biocrust cover (i.e., low vs. high), warming and rainfall reduction on metal content and bioavailability after 7 years of experimental manipulation (i.e., 2017) using linear models. We analysed the specific effects, compared to the ambient treatment, of rainfall exclusion, warming and

rainfall exclusion combined with warming on total and available metals in 2017 by linear regressions, separating high and low biocrust plots. We also used linear models to evaluate the univariate and interaction effects of biocrust cover, warming and rainfall manipulation on the change of all the response variables from 2010 to 2017. For linear models, we used the lm function from the *stats* package in R[71] and the results are shown in Supplementary Tables 1 and 2. Moreover, we used Permutational Analysis of Variance (PerMANOVA) to evaluate how total metal content and bioavailability (in terms of base cations and metallic micronutrients separately) changed from 2010 to 2017 in response to our treatments in a coordinated fashion, for which we used the adonis function from the *vegan* package[72] and the results are shown in Supplementary Tables 3 and 4. Results from the PerMANOVA were better visualised by carrying out principal component analyses using the rda function from vegan.

Finally, we sought to build a system-level understanding of the effects of initial biocrust cover, warming, rainfall reduction and soil pH, which is known to influence many soil properties and reactions, on total metal content and bioavailability using structural equation modelling (SEM)[73,74], for which we used the sem function of lavaan[75]. In our a priori conceptual models (Supplementary Fig. 4), treatments were predicted to directly affect soil pH and total metal content and availability; soil pH was, in turn, predicted to affect total metal content and availability; and total metal content was predicted to affect metal availability. Soil C was not included in our models as it was highly correlated with total nutrient contents and was thus deemed as redundant. We carried out separate SEMs for each metallic nutrient, both for total content and availability in 2017 and for the % of change from 2010 to 2017. Models were considered to have a good fit to our data when the $P$ values associated with the chi-square ($\chi^2$) and RMSEA statistics were >0.05.

**Reporting summary**. Further information on research design is available in the Nature Research Reporting Summary linked to this article.

## Data availability

The data from this work are available at https://figshare.com/s/4e36e5b55a49bc038cc4. Any other data not present in the article or supplementary files are available from the authors upon reasonable request.

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

## Acknowledgements

We thank Mónica Ladrón de Guevara, José Luis Quero and Sergio Asensio for their help with the monitoring and maintenance of the experiment and to Iván Frutos for his assistance with lab procedures. This work was funded by a Project supported by a 2018 Leonardo Grant for Researchers and Cultural Creators of the BBVA Foundation, by the research project CGL2016-78075-P (DINCOS), of the Spanish National Program of Scientific and Technical Research and by the European Research Council (ERC Grant agreements 242658 [BIOCOM] and 647038 [BIODESERT]). F.T.M. also acknowledges support from Generalitat Valenciana (CIDEGENT/2018/041).

## Author contributions

F.T.M. designed the experiment. E.M.-J, C.P., R.O.-H. and F.T.M designed the study. V.O., B.G., R.L. and F.T.M. carried out the field work. V.O., B.G., M.F., F.Z.E. and S.A.-H. prepared, processed and analysed the soil samples. E.M.-J., R.O.-H., M.F., C.P. and F.T.M. contributed to data analysis and interpretation. E.M.-J. drafted the manuscript and all the co-authors contributed significantly to the writing. All authors commented on and approved the final manuscript.

## Competing interests

The authors declare no competing interests.
