## [Peer Review File · Communications Biology]

Reviewers' comments:

Reviewer #1 (Remarks to the Author):

This manuscript reports the results of a long-term climate change experiment in arid drylands in Spain with respect to amounts and availability of metallic nutrients in the soil. The plots differed in their initial amount of biocrust cover (split into 32 "low" and 32 "high") and were assigned to warming (passive or control) and rainfall reduction (less or control) treatments in a factorial design. The main findings are that warming and rainfall reduction increase the availabilities of some metallic nutrients but a more well-developed biocrust limited these changes in availability. Overall, it is a strong study that addresses an important topic. The level of replication and duration of the study are impressive. The use of the literature is good. My major concerns are about referring to biocrust cover as a treatment when it is a pre-existing condition and some questions about which results are being referred to in some places (as the ones that appear to the ones referred to don't match what is being reported).

The biocrust factor is not an experimental treatment. It is an initial condition that was not created by the researchers. This is important for multiple reasons. First, unlike an experimental treatment randomly assigned to a plot, it is possible that some other variable is driving differences in both biocrust extent and metallic nutrients (throughout the experiment). Second, you cannot discount the possibility that it is differences in metallic nutrients in 2010 that caused the differences in biocrust extent. Third, the change in metallic nutrients may also depend on some other external factor. Everywhere that the manuscript reports results like "In the case of K, we also found a negative, direct effect of the presence of high initial biocrust cover" or "greater biocrust cover increased the availability of micronutrients through time" need to be rewritten to reflect that there is no clear causative relationship for biocrust extent. Likewise, statements in the discussion like "Biocrusts increased the abundance and/or availability of several metallic nutrients [175]" are not correct but rather should say something like "Higher levels of biocrusts were associated with greater abundance and/or availability of several metallic nutrients". Or "Biocrusts buffered against the strong increase in the availability of some metallic.. [187]" might become "Plots with higher initial biocrust cover were less sensitive to...".

Line 121-3 – the text says "Seven years of reduced rainfall combined with warmer temperatures increased the contents of available K, Mg, Na and Zn in soils under low, but not high, biocrust cover (Figs. 1 and SM1)" but Fig 1 is the amount in 2017 not the increase [and so has nothing to do with this statement] and the only significant effects in Table SM1 for these minerals are for the main effect of initial biocrust category. The text appears to refer to significant effects of the interaction of rainfall and temperature but the corresponding P-values are not significant (0.35 or greater) and so this statement is incorrect. Either refer to a significant result or remove this from the text. Related to this, if there was in fact such a significant interaction of warming and temperature, then I do not think that the SEM model structures used would be adequate as they do not include interactions in the effects of those 2 variables.

Fig 3 – are these values in 2017? The legend is not clear as "effects on" could mean changes or final values (and in this case the initial values were not the same for low vs. high biocrusts)

Line 117 – the text says "Warming consistently increased the content and availability of metallic macronutrients" but that variable does not appear in table SM4 – is this the "basic cations" in SM4? If so, the tables and text need to be made consistent.

Did the warming treatment affect rainfall input? I am curious because the clear sheets slope inwards over the plot. If so, how much? If so, add some discussion about this potentially contributing to

warming*rainfall interactions or part of the warming effect actually being a rainfall reduction effect.

Ref line 80 - format

Reviewer #2 (Remarks to the Author):

The manuscript reports experimental results on metallic nutrient availability in response to three factors (warming, rainfall, and biocrust cover) in a Mediterranean dryland ecosystem. The main results are that authors found increases in the availability of K, Mg, Zn and Na under warming and rainfall exclusion. The presence of a well-developed biocrust cover buffered these effects, most probably because its constituent species can uptake significant quantities of available forms of these metallic nutrients. The authors had carried out a manipulation experiment to assessing the effect of experimental warming, rainfall reduction (33-36% exclusion) and biocrust cover on soil metallic nutrient contents and availability. Here, the experiment was implemented based on a full factorial design.

The results presented in the manuscript seem interesting. The methods and conclusions well organized and clearly write at present version. I am very enjoying to read this interesting experimental manuscript. To summarize, I recommend it publish after a minor revision.

Some specific comments:

L80, the earlier literature should be cited here, Jones 1993 *Oikos*, which has proposed the concept of ecosystem engineers.

L238, I would suggest that authors present more detail information on temperature control? How did you increase 2-3 °C within your plots?

L499, the statistically significant levels should indicate in the Figure 1 even there is no significant difference.

EDITOR

*Thank you so much for your patience with the review process. Your manuscript entitled "Biocrusts buffer against the accumulation of soil metallic nutrients induced by seven years of warming and rainfall reduction in a dryland ecosystem" has now been seen by 2 referees. You will see from their comments below that they find your work of considerable interest, but have raised some relatively minor points. We are interested in the possibility of publishing your study in *Communications Biology*, but would like to consider your response to these concerns in the form of a revised manuscript before we make a final decision on publication.*

We therefore invite you to revise and resubmit your manuscript, taking into account the points raised by the referees. Please highlight all changes in the manuscript text file.

- 1. We would like to thank you for the opportunity to resubmit our work for further consideration in *Communications Biology*. We have revised the manuscript taking into account all the points raised by the reviewers, as detailed point-by-point below. As per your instructions, in the resubmission material, we have also included a Word file showing all the changes made in the original text. Additionally, we have made some small improvements in the text.**

REVIEWER 1

This manuscript reports the results of a long-term climate change experiment in arid drylands in Spain with respect to amounts and availability of metallic nutrients in the soil. The plots differed in their initial amount of biocrust cover (split into 32 "low" and 32 "high") and were assigned to warming (passive or control) and rainfall reduction (less or control) treatments in a factorial design. The main findings are that warming and rainfall reduction increase the availabilities of some metallic nutrients but a more well-developed biocrust limited these changes in availability. Overall, it is a strong study that addresses an important topic. The level of replication and duration of the study are impressive. The use of the literature is good. My major concerns are about referring to biocrust cover as a treatment when it is a pre-existing condition and some questions about which results are being referred to in some places (as the ones that appear to the ones referred to don't match what is being reported).

- 2. First of all, thank you for the time you devoted to our work and the positive comments. We have addressed all your concerns and revised the manuscript accordingly as described below.**

The biocrust factor is not an experimental treatment. It is an initial condition that was not created by the researchers. This is important for multiple reasons. First, unlike an experimental treatment randomly assigned to a plot, it is possible that some other variable is driving differences in both biocrust extent and metallic nutrients (throughout the experiment). Second, you cannot discount the possibility that it is differences in metallic nutrients in 2010 that caused the differences in biocrust extent. Third, the change in metallic

nutrients may also depend on some other external factor. Everywhere that the manuscript reports results like “In the case of K, we also found a negative, direct effect of the presence of high initial biocrust cover” or “greater biocrust cover increased the availability of micronutrients through time” need to be rewritten to reflect that there is no clear causative relationship for biocrust extent. Likewise, statements in the discussion like “Biocrusts increased the abundance and/or availability of several metallic nutrients [175]” are not correct but rather should say something like “Higher levels of biocrusts were associated with greater abundance and/or availability of several metallic nutrients”. Or “Biocrusts buffered against the strong increase in the availability of some metallic. [187]” might become “Plots with higher initial biocrust cover were less sensitive to...”.

- 3. The reviewer is right that biocrust cover is not a manipulative treatment but an initial experimental condition. Thus, we have carefully revised all the sentences in the text as suggested, to avoid the conclusion that there is a clear causal relationship between the initial biocrust cover and the dependent variables examined (soil metals) (for example Lines 142, 144, 188-9, 202, 209, 244-5).**

Line 121-3 – the text says “Seven years of reduced rainfall combined with warmer temperatures increased the contents of available K, Mg, Na and Zn in soils under low, but not high, biocrust cover (Figs. 1 and SM1)” but Fig 1 is the amount in 2017 not the increase [and so has nothing to do with this statement] and the only significant effects in Table SM1 for these minerals are for the main effect of initial biocrust category. The text appears to refer to significant effects of the interaction of rainfall and temperature but the corresponding P-values are not significant (0.35 or greater) and so this statement is incorrect. Either refer to a significant result or remove this from the text. Related to this, if there was in fact such a significant interaction of warming and temperature, then I do not think that the SEM model structures used would be adequate as they do not include interactions in the effects of those 2 variables.

- 4. The text in l. 121-123 refers to available contents and to Figure 1 and Figure SM1, not to total contents and Table SM1. We have slightly reworded the text to clarify these aspects. We have also indicated t and P values for significant contrasts with the ambient treatment as baseline (lines 123-130). The interaction effects between warming and rainfall on total and available contents were not significant (or marginally significant in the case of available Zn) (Tables SM1 and SM2), and for the sake of parsimony, we did not include such interactions in our SEMs.**

Fig 3 – are these values in 2017? The legend is not clear as “effects on” could mean changes or final values (and in this case the initial values were not the same for low vs. high biocrusts)

- 5. That is correct. Fig. 3 shows the effects of the treatments in 2017, seven years after the beginning of the experiment. We have revised the caption of Fig. 3 (and also that of Fig. SM5) accordingly to clarify this aspect.**

Line 117 – the text says “Warming consistently increased the content and availability of metallic macronutrients” but that variable does not appear in table SM4 – is this the “basic cations” in SM4? If so, the tables and text need to be made consistent.

- 6. The reviewer is correct that the terms “metallic macronutrients” and “basic cations” are being used interchangeably in our work. We have followed the reviewer’s suggestion and have consistently used the term “metallic macronutrients” (and not “basic cations”) throughout the revised manuscript for the sake of clarity.**

*Did the warming treatment affect rainfall input? I am curious because the clear sheets slope inwards over the plot. If so, how much? If so, add some discussion about this potentially contributing to warming*rainfall interactions or part of the warming effect actually being a rainfall reduction effect.*

- 7. The effects of the warming treatment on soil moisture were assessed in a previous study from the same experiment (Maestre et al., 2013). We found that the sloping sides of the chambers decrease soil moisture by ~1.5%. As suggested, we have reported this effect in the revised manuscript (Lines 269-271).**

Ref line 80 – format

- 8. This revision has been made.**

REVIEWER 2

The manuscript reports experimental results on metallic nutrient availability in response to three factors (warming, rainfall, and biocrust cover) in a Mediterranean dryland ecosystem. The main results are that authors found increases in the availability of K, Mg, Zn and Na under warming and rainfall exclusion. The presence of a well-developed biocrust cover buffered these effects, most probably because its constituent species can uptake significant quantities of available forms of these metallic nutrients. The authors had carried out a manipulation experiment to assessing the effect of experimental warming, rainfall reduction (33-36% exclusion) and biocrust cover on soil metallic nutrient contents and availability. Here, the experiment was implemented based on a full factorial design.

The results presented in the manuscript seem interesting. The methods and conclusions well organized and clearly write at present version. I am very enjoying to read this interesting experimental manuscript. To summarize, I recommend it publish after a minor revision.

- 9. We thank reviewer #2 for these very positive comments.**

Some specific comments:

L80, the earlier literature should be cited here, Jones 1993 Oikos, which has proposed the concept of ecosystem engineers.

10. We have cited this paper in the revised manuscript, as it is indeed relevant to our work (new ref. 20). In addition, we introduced the concept in the abstract (Line 35).

L238, I would suggest that authors present more detail information on temperature control? How did you increase 2-3 °C within your plots?

11. As mentioned in the original manuscript, we increased air temperatures on the plots by 2-3 °C just with open top chambers. We have clarified this aspect and provided some further information in the revised manuscript.

L499, the statistically significant levels should indicate in the Figure 1 even there is no significant difference.

12. We have indicated t and P values in the revised text for contrasts with the ambient treatment as baseline (Lines 123-125).

References

- Jones, Clive G., John H. Lawton, and Moshe Shachak. Organisms as Ecosystem Engineers, *Oikos* 69 (1994) 373-386
- Maestre, F. T. et al. Changes in biocrust cover drive carbon cycle responses to climate change in drylands. *Glob. Chang. Biol.* 19 (2013) 3835–3847

REVIEWERS' COMMENTS:

Reviewer #2 (Remarks to the Author):

The authors have addressed my concerns in the revision. I am happy to recommend it at present version.